# Position: Deep Learning is Not So Mysterious or Different

**Andrew Gordon Wilson** [1]

## Abstract

Deep neural networks are often seen as different from other model classes by defying conventional notions of generalization. Popular examples of anomalous generalization behaviour include benign overfitting, double descent, and the success of overparametrization. This position paper argues that these phenomena are not distinct to neural networks, or particularly mysterious. Moreover, this generalization behaviour can be intuitively understood, and rigorously characterized, using long-standing generalization frameworks such as PAC-Bayes and countable hypothesis bounds. We present *soft inductive biases* as a key unifying principle in explaining these phenomena: rather than restricting the hypothesis space to avoid overfitting, embrace a flexible hypothesis space, with a soft preference for simpler solutions that are consistent with the data. This principle can be encoded in many model classes, and thus deep learning is not as mysterious or different from other model classes as it might seem. However, we also highlight how deep learning is relatively distinct in other ways, such as its ability for representation learning, phenomena such as mode connectivity, and its relative universality.

## 1. Introduction

*"The textbooks must be re-written!"*

Deep neural networks are often considered mysterious and different from other model classes, with behaviour that can defy the conventional wisdom about generalization. When asked what makes deep learning different, it is common to point to phenomena such as *overparametrization*, *double descent*, and *benign overfitting* (Zhang et al., 2021; Nakkiran et al., 2020; Belkin et al., 2019; Shazeer et al., 2017).

**Our position is that none of these phenomena are dis-**
**tinct to neural networks, or particularly mysterious.**
Moreover, while some generalization frameworks such as VC dimension (Vapnik, 1998) and Rademacher complexity (Bartlett & Mendelson, 2002) do not explain these phenomena, they are **formally described by other long-standing frameworks** such as PAC-Bayes (McAllester, 1999; Catoni, 2007; Dziugaite & Roy, 2017), and even simple countable hypothesis generalization bounds (Valiant, 1984; Shalev-Shwartz & Ben-David, 2014; Lotfi et al., 2024a). *In other words, understanding deep learning does not require rethinking generalization, and it never did.*

We are *not* aiming to argue that deep learning is fully understood, to comprehensively survey works on understanding deep learning phenomena, or to assign historical priority to any work for explaining some phenomenon. We are also not claiming to be the first to note that any of these phenomena can be reproduced using other model classes. In fact, we want to make clear that there has been significant progress in understanding what is often perceived as mysterious generalization behaviour in deep learning, and contrary to common belief, much of this behaviour applies outside of deep learning and can be formally explained using frameworks that have existed for decades. The textbooks wouldn't need to be re-written had they paid attention to what was already known about generalization, decades ago! Instead, we need to bridge communities, and acknowledge progress.

Indeed, we will aim to introduce the *simplest* examples possible, often basic linear models, to replicate these phenomena and explain the intuition behind them. The hope is that by relying on particularly simple examples, we can drive home the point that these generalization behaviours are hardly distinct to neural networks and can in fact be understood with basic principles. For example, in Figure 1, we show that benign overfitting and double descent can be reproduced and explained with simple linear models.

We will also treat all of these phenomena collectively, through a unifying notion of *soft inductive biases*. While *inductive biases* are often thought of as *restriction biases* — constraining the size of a hypothesis space for improved data efficiency and generalization — there is no need for restriction biases. Instead, we can embrace an arbitrarily flexible hypothesis space, combined with soft biases that express a preference for certain solutions over others, without entirely

---

[1]New York University. Correspondence to: Andrew Gordon Wilson <andrewgw@cims.nyu.edu>.

*Proceedings of the 42nd International Conference on Machine Learning*, Vancouver, Canada. PMLR 267, 2025.

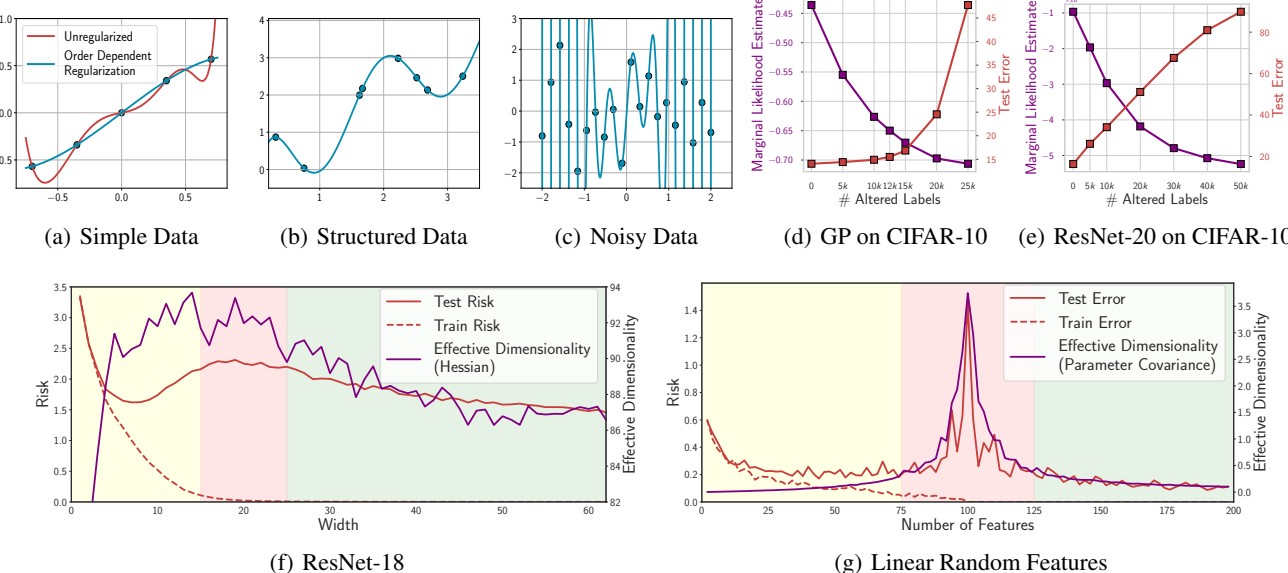

(a) Simple Data     (b) Structured Data     (c) Noisy Data     (d) GP on CIFAR-10     (e) ResNet-20 on CIFAR-10

(f) ResNet-18          (g) Linear Random Features

*Figure 1.* **Generalization phenomena associated with deep learning can be reproduced with simple linear models and understood. Top: Benign Overfitting.** A $150th$ order polynomial with order-dependent regularization reasonably describes (a) simple and (b) complex structured data, while also being able to perfectly fit (c) pure noise. (d) A Gaussian process exactly reproduces the CIFAR-10 results in Zhang et al. (2016), perfectly fitting noisy labels, but still achieving reasonable generalization. Moreover, for both the GP and (e) ResNet, the marginal likelihood, directly corresponding to PAC-Bayes bounds (Germain et al., 2016), decreases with more altered labels, as in Wilson & Izmailov (2020). **Bottom: Double Descent.** Both the (f) ResNet and (g) linear random feature model display double descent, with effective dimensionality closely tracking the second descent in the low training loss regime as in Maddox et al. (2020).

ruling out any solution, as illustrated in Figure 3. Frameworks such as PAC-Bayes embody this view of inductive biases, capable of producing non-vacuous generalization bounds on models with even billions of parameters, as long as these models have a prior preference for certain solutions over others (Lotfi et al., 2024b). Broadly speaking, a large hypothesis space, combined with a preference for simple solutions, provides a provably useful recipe for good performance, as in Figure 2.

There are also other phenomena of recent interest, such as *scaling laws* and *grokking*, which are not our focus, because these are not typically treated as inconsistent with generalization theory, or distinct to neural networks. However, we note the PAC-Bayes and countable hypothesis generalization frameworks of Section 3 are also consistent with LLMs, and even Chinchilla scaling laws (Hoffmann et al., 2022; Finzi et al., 2025). Moreover, deep learning of course *is* different in other ways. In Appendix A, we discuss relatively distinctive features of deep neural networks, such as representation learning, mode connectivity, and broadly successful in-context learning.

We open with a discussion of soft inductive biases in Section 2, which provide a unifying intuition throughout the paper. We then briefly introduce several general frameworks and definitions in Section 3, preliminaries through which

we examine generalization phenomena in the next sections. Throughout the paper, we particularly contrast PAC-Bayes and the countable hypothesis frameworks in Section 3.1, which do characterize these generalization phenomena, with other generalization frameworks such as Rademacher complexity and VC dimension in Section 3.3 which do not.

## 2. Soft Inductive Biases

We often think of inductive biases as *restriction biases*: constraints to the hypothesis space aligned with a problem of interest. In other words, there are many settings of parameters that may fit the data and provide poor generalization, so restrict the hypothesis space to settings of parameters that are more likely to provide good generalization for the problem we are considering. Moreover, since the hypothesis space is smaller, it will become more quickly constrained by the data, since we have fewer solutions to "rule out" with the addition of new data points. Convolutional neural networks provide a canonical example: we start from an MLP, remove parameters, and enforce parameter sharing, to provide a hard constraint for locality and translation equivariance.

But restriction biases are not only unnecessary, they are arguably undesirable. We want to support any solution that could describe the data, which means embracing a flexible hypothesis space. For example, we may suspect the data are

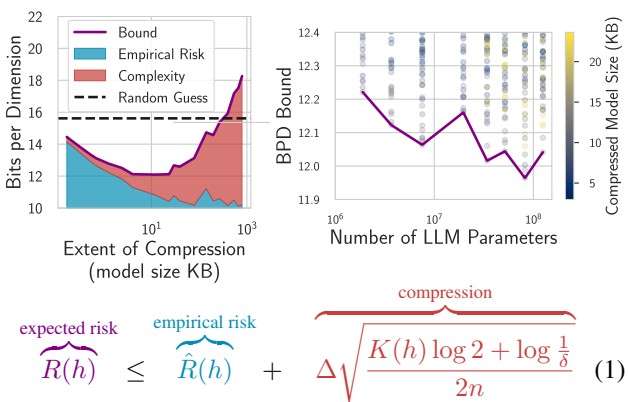

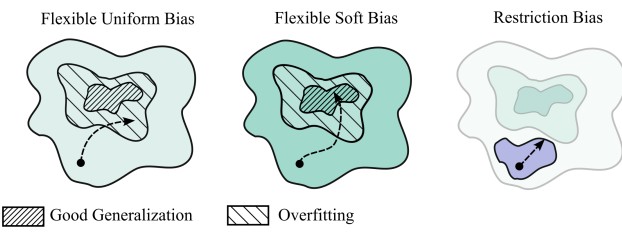

*Figure 4.* **Achieving good generalization with soft inductive biases. Left:** A large hypothesis space, but no preference amongst solutions that provide the same fit to the data. Therefore, training will often lead to overfit solutions that generalize poorly. **Middle:** Soft inductive biases guide training towards good generalization by representing a flexible hypothesis space in combination with preferences between solutions, represented by different shades. **Right:** Restricting the hypothesis space can help prevent overfitting by only considering solutions that have certain desirable properties. However, by limiting expressiveness, the model cannot capture the nuances of reality, hindering generalization.

$$\overbrace{R(h)}^{\text{expected risk}} \;\leq\; \overbrace{\hat{R}(h)}^{\text{empirical risk}} \;+\; \Delta\overbrace{\sqrt{\frac{K(h)\log 2 + \log\frac{1}{\delta}}{2n}}}^{\text{compression}} \quad (1)$$

*Figure 2.* **Generalization phenomena can be formally characterized by generalization bounds.** Generalization can be upper bounded by the empirical risk and compressibility of a hypothesis $h$, as in Section 3.1. The compressibility, formalized in terms of Kolmogorov complexity $K(h)$, can be further upper bounded by a model's filesize. Large models fit the data well, and can be effectively compressed to small filesizes. Unlike Rademacher complexity, these bounds do not penalize a model for having a hypothesis space $\mathcal{H}$ that can fit noise, and describe benign overfitting, double descent, and overparametrization. They can even provide non-vacuous bounds on LLMs, as in Lotfi et al. (2024a) above.

We refer to the general idea of having a preference for certain solutions over others, even if they fit the data equally well, as a *soft inductive bias*. We contrast soft biases with more standard restriction biases, which instead place hard constraints on the hypothesis space. We illustrate the concept of soft inductive biases in Figure 3, and show how soft inductive biases influence the training process in Figure 4. Regularization, as well as Bayesian priors over model parameters, provide mechanisms for creating soft inductive biases. However, regularization is not typically used to relax architectural constraints, and as we will see, soft biases are more general, and can be induced by the architecture.

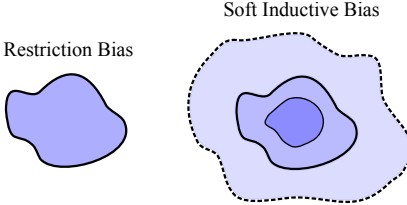

*Figure 3.* **Soft inductive biases enable flexible hypothesis spaces without overfitting.** In this conceptualization, we enlarge the hypothesis space with hypotheses that have lower preference in lighter blue, rather than restricting them entirely. There are many ways to implement soft inductive biases. Rather than use a low order polynomial, use a high order polynomial with order-dependent regularization. Alternatively, rather than restrict a model to translation equivariance, have a preference for invariances through a compression bias (e.g., a transformer, or RPP with ConvNet bias). Overparametrization is yet another way to implement a soft bias.

As a running example, consider a large polynomial, but where we regularize the higher order coefficients more than the lower order coefficients. In other words, we fit the data with $f(x,w) = \sum_{j=0}^{J} w_j x^j$ and we have a regularizer on $w_j$ that increases in strength with $j$. Finally, we have a data fit term that is formed from a likelihood involving $f(x,w)$, $p(y|f(x,w))$. So our total loss is:

$$\text{Loss} = \text{data fit} + \text{order dependent complexity penalty}$$

which, for example, could take the form $\mathcal{L}(w) = -\log p(y|f(x,w)) + \sum_j \gamma^j w_j^2$, $\gamma > 1$. For classification, the observation model $p(y_i|f(x_i,w)) = \text{softmax}(f(x_i,w))$ would give rise to cross-entropy for $-\log p(y|f(x,w))$. In regression, $p(y_i|f(x_i,w)) = \mathcal{N}(f(x_i,w),\sigma^2)$ would give rise to the squared error data fit, divided by $1/(2\sigma^2)$.

If we take the order of the polynomial $J$ to be large, then we have a flexible model. But the model has a simplicity bias: due to the order dependent complexity penalty, it will try to fit the data using the lower order terms as much as possible, and then only use the higher order terms if needed. For

only approximately translation equivariant. We can instead bias the model towards translation equivariance without any hard constraint. A naive way to provide a soft ConvNet bias would be to start with an MLP, and then introduce a regularizer that penalizes both the norms of any parameters that do not exist in a ConvNet, and the distance between any parameters that would otherwise be shared in a ConvNet. We can control this bias through the strength of the regularization. *Residual pathway priors* provide a more practical and general mechanism for turning hard architectural constraints into soft inductive biases (Finzi et al., 2021).

example, imagine a simple 1d regression problem, where the data fall onto a straight line. For large $J$, there are many settings of the coefficients $\{w_j\}$ that will perfectly fit the data. But the model will prefer the simple straight line fit with $w_j = 0$ for $j \geq 2$ because it's consistent with the data and incurs the lowest penalty, as in Figure 1 (top left). In effect, we have relaxed the hard restriction bias of a low-order polynomial, and turned it into a soft inductive bias. Such a model is also effective for any size of training set: on small datasets it is competitive with models that have hard constraints, on large datasets it is competitive with relatively unconstrained models, as depicted in Figure 6.

While $\ell_2$ and $\ell_1$ (or Lasso) regularization is standard practice, it is not used as a prescription for building models of arbitrary size. The idea of order-dependent regularization is less known. Rasmussen & Ghahramani (2000) show the Bayesian marginal likelihood (evidence), the probability of generating the training data from the prior, favours higher-order Fourier models with a similar order-dependent parameter prior. A prior over parameters $p(w)$ induces a prior over functions $p(f(x, w))$, and from the Bayesian perspective it is this prior over functions that controls the generalization properties of the model (Wilson & Izmailov, 2020). An order-dependent prior gives rise to a prior over functions that may likely generate the data, even for high-order models. On the other hand, six years after Rasmussen & Ghahramani (2000), the canonical textbook Bishop (2006) argues in Chapter 3, page 168, that the marginal likelihood is aligned with conventional notions of model selection, *precisely because it chooses a polynomial of intermediate order, rather than a small or large polynomial*. In actuality, this textbook result is simply an artifact of a bad prior: it uses an isotropic parameter prior (analogous to $\ell_2$ regularization), and a high-order polynomial with an isotropic parameter prior is unlikely to generate the data. Had Bishop (2006) chosen an order-dependent prior, the marginal likelihood could have preferred an arbitrarily high-order model.

In Residual Pathway Priors (RPP) (Finzi et al., 2021), it was shown that a soft bias for equivariance constraints was often as effective as a model that had been *perfectly* constrained for a given problem. For example, a soft bias for rotation equivariance would work as well as a rotationally equivariant model for molecules, which are rotation invariant. After exposure to only a very small amount of data, the soft bias would converge to near-perfect rotation equivariance, since the model is encouraged (but not constrained) to represent the data with symmetries, and it can do so exactly, even with a small amount of data. Moreover, in cases where the data only contained an approximate symmetry, or no symmetry at all, the RPP approach would significantly outperform a model with hard symmetry constraints.

Surprisingly, vision transformers after training can be even more translation equivariant than convolutional neural networks (Gruver et al., 2023)! This finding may seem impossible, as ConvNets are architecturally constrained to be translation equivariant. However, in practice equivariance is broken by aliasing artifacts. Equivariance symmetries provide a mechanism for compressing the data, and as we will discuss in later sections, transformers have a soft inductive bias for compression.

It is our view that *soft* inductive biases, rather than constraining the hypothesis space, are a key prescription for building intelligent systems.

## 3. Generalization Frameworks

We have so far argued that we intuitively want to embrace a flexible hypothesis space, because it represents our honest beliefs that real-world data will have sophisticated structure. But in order to have good generalization, we must have a prior bias towards certain types of solutions, even if we are allowing for any type of solution. While the generalization phenomena we discuss defy some conventional wisdom around overfitting and notions of generalization such as Rademacher complexity, as argued in Zhang et al. (2016; 2021), they are entirely aligned with this intuition.

It turns out these phenomena are also **formally characterized** by generalization frameworks that have existed for many decades, including PAC-Bayes (McAllester, 1999; Guedj, 2019; Alquier et al., 2024) and simple countable hypothesis bounds (Valiant, 1984; Shalev-Shwartz & Ben-David, 2014). We introduce these frameworks in Section 3.1. We then define *effective dimensionality* in Section 3.2 which we will return to later in the paper for intuition. Finally we introduce frameworks that do *not* describe these phenomena in Section 3.3, but have greatly impacted the conventional wisdom in thinking about generalization.

This section briefly introduces some definitions and generalization frameworks — preliminaries through which we will examine generalization phenomena in later sections.

### 3.1. PAC-Bayes and countable hypothesis bounds

PAC-Bayes and countable hypothesis bounds provide a compelling approach for large and even overparametrized models, since they are focused on which hypotheses are *likely*, rather than merely the size of the hypothesis space (Catoni, 2007; Shalev-Shwartz & Ben-David, 2014; Dziugaite & Roy, 2017; Arora et al., 2018b; Pérez-Ortiz et al., 2021; Lotfi et al., 2022a). They harmonize with the notion of *soft* inductive biases in Section 2, which provide a mechanism for achieving good generalization with an arbitrarily large hypothesis space combined with preferences for certain solutions over others independently of their fit to the data.

**Theorem 3.1** (Countable Hypothesis Bound). *Consider a bounded risk $R(h, x) \in [a, a + \Delta]$, and a countable hypothesis space $h \in \mathcal{H}$ for which we have a prior $P(h)$. Let the empirical risk $\hat{R}(h) = \frac{1}{n} \sum_{i=1}^{n} R(h, x_i)$ be a sum over independent random variables $R(h, x_i)$ for a fixed hypothesis $h$. Let $R(h) = \mathbb{E}[\hat{R}(h)]$ be the expected risk. Then, with probability at least $1 - \delta$,*

$$R(h) \leq \hat{R}(h) + \Delta \sqrt{\frac{\log \frac{1}{P(h)} + \log \frac{1}{\delta}}{2n}}. \qquad (2)$$

This bound is related to the finite hypothesis bound, but includes a prior $P(h)$ and a *countable* rather than finite hypothesis space (Ch 7.3, Shalev-Shwartz & Ben-David, 2014). We can think of the prior as a weighting function that weights certain hypotheses more highly than others. Importantly, we can use any prior to evaluate the bound: it need not have generated the true hypothesis for the data, contain the true hypothesis, or even be used by the model that is trained to find some hypothesis $h^*$. If the model uses a prior quite different from the prior used to evaluate Eq. (2), then the bound will simply become loose. We include an elementary proof of this bound in Appendix D.

We can derive informative bounds through a Solomonoff prior $P(h) = 2^{-K(h|A)}/Z$ (Solomonoff, 1964), where $K$ is the prefix-free Kolmogorov complexity of $h$ taking as input model architecture $A$, and the normalizing constant $Z \leq 1$ by the Kraft inequality (Kraft, 1949). Substituting this prior into Eq. (2) yields Eq. (1). The prefix-free *Kolmogorov complexity* of hypothesis $h$, $K(h)$, is the length of the shortest program that produces $h$ for a fixed programming language (Kolmogorov, 1963). While we cannot compute the *shortest* program, we can absorb the architecture and any constant not determined by the data into the prior, by working with $K(h|A)$. We can then convert from the prefix-free to standard Kolmogorov complexity, to compute the upper bound

$$\log 1/P(h) \leq K(h|A) \log 2 \leq C(h) \log 2 + 2 \log C(h) \qquad (3)$$

where $C(h)$ is the number of bits required to represent hypothesis $h$ using some pre-specified coding. Therefore even large models with many parameters that represent hypotheses with a low empirical risk and a small compressed size can achieve strong generalization guarantees.

*PAC-Bayes bounds* can further reduce the number of bits required from $\log_2 \frac{1}{P(h)}$ to $\mathbb{KL}(Q \parallel P)$ by considering a distribution of desirable solutions $Q$. If we are agnostic to the specific element of $Q$ we sample, we can recover bits that could then be used to encode a different message. Since PAC-Bayes bounds with a point-mass posterior $Q$ can recover a bound similar to Eq. (2) (Lotfi et al., 2022b), we will sometimes refer to both bounds as PAC-Bayes. We also

note that *marginal likelihood*, which is the probability of generating the training data from the model prior, directly corresponds to a PAC-Bayes bound (Germain et al., 2016).

These generalization frameworks have been adapted to provide *non-vacuous generalization guarantees on models that have millions, or even billions, of parameters*. They apply to deterministically trained models, and have also been adapted to LLMs, to accommodate the unbounded bits-per-dimension (nats-per-token) loss, stochastic training, and dependence across tokens (Lotfi et al., 2023; 2024b; Finzi et al., 2025). Moreover, *computing these bounds is straightforward*. For example: (i) train a model to find hypothesis $h^*$, using any optimizer; (ii) measure the empirical risk $\hat{R}(h^*)$ (e.g., training loss); (iii) measure the filesize of the stored model for $C(h^*)$; (iv) substitute Eq. (3) into Eq. (1).

In words, we can interpret these generalization bounds as:

Expected Risk $\leq$ Empirical Risk $+$ Model Compressibility

where compressibility provides a formalization of complexity. In Figure 2, adapted from Lotfi et al. (2023), we visualize how each term contributes to the bound. This representation of the bounds also provides a *prescription* for building general-purpose learners: combine a flexible hypothesis space with a bias for low Kolmogorov complexity. A flexible model will be able to achieve low empirical risk (training loss) on a wide variety of datasets. Being able to compress these models will then provably lead to good generalization. Goldblum et al. (2024) show that neural networks, especially large transformers, tend to be biased towards low Kolmogorov complexity, and so is the distribution over real-world data. For this reason, a single model can generalize well over many real-world problems.

Indeed, even within a *maximally flexible hypothesis space* consisting of all possible programs, if we choose a hypothesis that fits the data well and has low complexity then we will be guaranteed to generalize by the countable hypothesis bound in Eq. (2). We can relate this insight to *Solomonoff induction*, which provides a maximally overparametrized procedure, with no limit on the complexity or number of parameters a hypothesis can have, but formalizes an ideal learning system (Solomonoff, 1964; Hutter, 2000). By assigning exponentially higher weights to simpler (shorter) programs, Solomonoff induction ensures that even though the hypothesis space is enormous, the chosen hypothesis will be simple if it fits the data well.

In general, there are **common misconceptions about PAC-Bayes and countable-hypothesis bounds.** For example, they do apply to models with deterministic parameters, rather than only distributions over parameters. Moreover, recent bounds become tighter, not looser, with larger models. We discuss several misconceptions in Appendix B. It is also worth noting that these bounds are not only non-vacuous for

large neural networks, but also can be surprisingly tight. For example, Lotfi et al. (2022a) upper bound the classification error of a model with millions of parameters on CIFAR-10 at 16.6% with at least 95% probability, which is fairly respectable performance on this benchmark.

### 3.2. Effective Dimensionality

Effective dimensionality provides a useful intuition for explaining generalization phenomena. The *effective dimensionality* of a matrix $A$ is $N_{\text{eff}}(A) = \sum_i \frac{\lambda_i}{\lambda_i + \alpha}$, where $\lambda_i$ are the eigenvalues of $A$, and $\alpha$ is a free parameter. The effective dimensionality measures the number of relatively large eigenvalues. The effective dimensionality of the Hessian of the loss, evaluated for parameters $w$, measures the number of sharp directions in the loss landscape — the number of parameters determined from the data. Solutions with lower effective dimensionality are *flatter*, meaning that the associated parameters can be perturbed without significantly increasing the loss. We have a mechanistic understanding of why flatness can lead to better generalization: flatter solutions are more compressible, have better Occam factors, tend to lead to wider decision boundaries, and tighter generalization bounds (Hinton & Van Camp, 1993; Hochreiter & Schmidhuber, 1997; MacKay, 2003; Keskar et al., 2016; Izmailov et al., 2018; Foret et al., 2020; Maddox et al., 2020). We will return to effective dimensionality for intuition when discussing generalization phenomena.

### 3.3. Other Generalization Frameworks

*Rademacher complexity* (Bartlett & Mendelson, 2002) exactly measures the ability for a model to fit uniform $\{+1, -1\}$ random noise. Similarly, the *VC dimension* (Vapnik et al., 1994) measures the largest integer $d$ such that the hypothesis space $\mathcal{H}$ can fit ("shatter") any set of $d$ points with $\{+1, -1\}$ labels. The *fat-shattering dimension* (Alon et al., 1997) $\text{fat}_\gamma(\mathcal{H})$ refines the VC dimension to fitting ("shattering") labels by some margin $\gamma$. Unlike PAC-Bayes, all of these frameworks penalize the *size* of the overall hypothesis space $\mathcal{H}$, suggesting a prescription for *restriction biases*, rather than the *soft inductive biases* of Section 2. We discuss these frameworks further in Appendix C, with a comparative summary in Table 1.

## 4. Benign Overfitting

*Benign overfitting* describes the ability for a model to fit noise with no loss, but still generalize well on structured data. It shows that a model can be *capable* of overfitting data, but won't tend to overfit structured data. The paper *understanding deep learning requires re-thinking generalization* (Zhang et al., 2016) drew significant attention to this phenomenon by showing that convolutional neural networks could fit images with random labels, but generalize well on

structured image recognition problems such as CIFAR. The result was presented as contradicting what we know about generalization, based on frameworks such as VC dimension and Rademacher complexity, and distinct to neural networks. The authors conclude with the claim: *"We argue that we have yet to discover a precise formal measure under which these enormous models are simple."* Five years later, the authors maintain the same position, with an extended paper entitled *understanding deep learning (still) requires re-thinking generalization* (Zhang et al., 2021). Similarly, Bartlett et al. (2020) note *"the phenomenon of benign overfitting is one of the key mysteries uncovered by deep learning methodology: deep neural networks seem to predict well, even with a perfect fit to noisy training data."*

However, benign overfitting behaviour can be reproduced with other model classes, can be understood intuitively, and is described by rigorous frameworks for characterizing generalization that have existed for decades.

**Intuition.** Intuitively, in order to reproduce benign overfitting, we just need a flexible hypothesis space, combined with a loss function that demands we fit the data, and a simplicity bias: amongst solutions that are consistent with the data (i.e., fit the data perfectly), the simpler ones are preferred. For a moment, consider regression, and the simple polynomial model with order-dependent regularization in Section 2. In our likelihood, we will drive $\sigma$ to a small value, so the model will prioritize fitting the data (squared error is multiplied by a large number). However, the model strongly prefers using the lower order terms, since the norms of coefficients are increasingly penalized with the order of the coefficient. Simple structured data will be fit with simple structured compressible functions that will generalize, but the model will adapt its complexity as needed to fit the data, including pure noise, as shown in Figure 1 (top). In other words, if understanding deep learning requires rethinking generalization, then understanding this simple polynomial does too, *for this polynomial exhibits benign overfitting!*

**Formal generalization frameworks.** Benign overfitting is also characterized by PAC-Bayes and countable hypothesis bounds, which are formal and long-standing frameworks for characterizing generalization. We can evaluate these bounds for neural networks that exhibit benign overfitting, providing non-vacuous generalization guarantees (Dziugaite & Roy, 2017; Zhou et al., 2018; Lotfi et al., 2022a). Moreover, as we describe in Section 3, these generalization frameworks can precisely define how large neural networks are simple, through Kolmogorov complexity. In fact, larger neural networks often have an even stronger bias for low Kolmogorov complexity solutions (Goldblum et al., 2024).

**Mix of signal and noise.** The ability to fit a mix of signal and noise, but still achieve respectable generalization, can

also be reproduced and is characterized by the generalization frameworks in Section 3.1. In particular, we can *exactly* reproduce the mixed noisy-label experiment in Zhang et al. (2021) for CIFAR-10 in Figure 1(d)(e), following Wilson & Izmailov (2020). Here a Gaussian process (GP) is fit to CIFAR-10 with no training error but increasing numbers of altered labels. Generalization is reasonable, and steadily degrades with increasing numbers of altered labels. Importantly, both the GP and ResNet marginal likelihoods decrease, and the marginal likelihood directly aligns with PAC-Bayes generalization bounds (Germain et al., 2016).

**Research on benign overfitting.** There is by now a large body of work studying and reproducing benign overfitting with other model classes. Yet the conventional wisdom of benign overfitting as a mysterious and deep learning specific phenomenon, one that *still* requires rethinking generalization, persists. It is not our intention, nor would it be possible, to cover all of this work here, but we note some of the key developments. Dziugaite & Roy (2017) show non-vacuous and vacuous PAC-Bayes bounds for neural networks trained on structured and noisy MNIST, respectively. Smith & Le (2018) demonstrate benign overfitting for logistic regression on MNIST, interpreting the results using Bayesian Occam factors (MacKay, 2003). Several studies analyze two-layer networks (e.g., Cao et al., 2022; Kou et al., 2023). Wilson & Izmailov (2020) exactly reproduce the experiments in Zhang et al. (2016) with Gaussian processes and Bayesian neural networks, and explain the results using marginal likelihood. Bartlett et al. (2020) show that linear regression models can reproduce benign overfitting. They understand this phenomenon by studying the rank of the data covariance matrix, and minimum-norm least squares solutions.

**Conclusion.** PAC-Bayes and the countable hypothesis bounds provide a "precise formal measure under which these enormous models are simple", while Rademacher complexity and VC dimension do not. Moreover, this generalization behaviour is intuitively understandable from the perspective of soft inductive biases, embracing a flexible hypothesis space combined with a compression bias.

## 5. Overparametrization

Now that we have covered soft biases, and benign overfitting, it is likely becoming increasingly intuitive that a model with many parameters will not necessarily overfit the data. Parameter counting, in general, is a poor proxy for model complexity. Indeed, before the resurgence of deep learning in 2012, it was becoming commonplace to embrace models with many parameters: *"it is now common practice for Bayesians to fit models that have more parameters than the number of data points..."* (MacKay, 1995).

We are not interested in the parameters in isolation, but rather how the parameters control the properties of the *func-*

*tions* we use to fit the data. We have already seen how arbitrarily large polynomials do not overfit the data, as long as they have a simplicity bias. Gaussian processes also provide compelling examples. A GP with an RBF kernel can be derived from an infinite sum of densely dispersed radial basis functions $\phi_i$: $f(x, w) = \sum_{i=1}^{\infty} w_i \phi_i(x)$ (MacKay, 1998). Similarly, using central limit theorem arguments, we can derive GP kernels corresponding to *infinite* single and multi-layer neural networks (Neal, 1996b; Lee et al., 2017; Matthews et al., 2018) (the first of these being an infamous NeurIPS rejection!). Indeed, GPs are typically more flexible than any standard neural network, but often have their *strongest* performance relative to other model classes on *small* datasets, due a strong (but soft) simplicity bias.

### 5.1. Is the success of overparametrization surprising?

There is seemingly little consensus on whether *overparametrization* is in fact surprising. On the one hand, it is known and understood within certain circles that models with an arbitrarily large number of parameters can generalize; indeed, pursuing the limits of large models has been a guiding principle in non-parametrics for decades (e.g., MacKay, 1995; Neal, 1996a; Rasmussen, 2000; Rasmussen & Ghahramani, 2000; Beal et al., 2001; Rasmussen & Ghahramani, 2002; Griffiths & Ghahramani, 2005; Williams & Rasmussen, 2006). At the same time, overparametrization has been a defining feature of neural networks. And many papers, especially theory papers, open by exclaiming surprise that deep neural networks can generalize given that they have more parameters than datapoints, particularly in light of benign overfitting: e.g., *"A mystery about deep nets is that they generalize despite having far more parameters than the number of training samples..."* (Arora et al., 2018a). Moreover, many generalization bounds also become increasingly loose, and eventually vacuous, as we increase the number of parameters (Jiang et al., 2019).

However, more recently, there have also been generalization bounds that become *tighter* as we increase the number of parameters (Lotfi et al., 2022a; 2024b; Finzi et al., 2025). While LLMs are in many cases not overparametrized, parameter counting is prevalent. And presentations of *double descent* (Section 6) are often based on parameter counting.

### 5.2. Why does increasing parameters help performance?

There are two reasons, flexibility and compression. We have discussed how models with high flexibility and a compression bias will provably provide good generalization (Section 3). Increasing the number of parameters in a neural network straightforwardly increases its flexibility. Perhaps more surprisingly, increasing the number of parameters also increases a compression bias: that is, *models with more parameters can be stored with less total memory after training than models with fewer parameters after training.*

Maddox et al. (2020) found that larger models after training had fewer *effective parameters* than smaller models, by measuring *effective dimensionality* of the Hessian (Section 3.2). In more recent work, Goldblum et al. (2024) also show that larger language models have a stronger simplicity bias, which is important for good in-context learning across multiple different settings and modalities.

But why do larger models appear to have a stronger compression bias? While this is a fascinating open question, there are some clues and intuitions. Bartlett et al. (2020) show that overparametrized least-squares models increasingly favour small-norm solutions with low effective rank (more in Section 6). As we increase the number of parameters, we can also exponentially increase the *volume* of flat solutions in the loss landscape, making them more easily accessible (Huang et al., 2019), which is empirically supported by larger models having smaller effective dimensionality (Maddox et al., 2020). This also helps explain why the *implicit biases of stochastic optimization, contrary to common belief, are not necessary for generalization in deep learning*: even though some parameter settings overfit the data, they are vastly outnumbered in volume by the parameter settings that fit the data well and also generalize well. Indeed, Geiping et al. (2021) found that full-batch gradient descent could perform nearly as well as SGD for training large residual networks, and Chiang et al. (2022) further showed that even *guess and check* — randomly sampling parameter vectors and stopping once a low-loss solution was found — can provide competitive generalization with stochastic training.

There is often a perceived tension between flexibility and inductive biases, with the assumption that more flexible models must have weaker inductive biases. But as we have seen, the larger and more flexible models often have *stronger* inductive biases, which we illustrate in Figure 7.

## 6. Double Descent

*Double descent* typically refers to generalization error (or loss) that decreases, then increases, then again decreases, with increases in the number of model parameters. The training loss is typically close to zero near the beginning of the second descent. The first decrease and then increase corresponds to a "classical regime", where the model initially captures more useful structure in the data, improving generalization, but then begins to overfit the data. The second descent, which gives rise to the name "double descent", is referred to as the "modern interpolating regime".

Double descent was introduced to the modern machine learning community by Belkin et al. (2019), and prominently studied for deep neural networks in Nakkiran et al. (2020). It is often considered one of the great mysteries of deep learning, with the second descent challenging the conven-

tional wisdom around generalization. If increasing model flexibility is leading to overfitting in the classical regime, how can further increasing flexibility alleviate overfitting? Belkin et al. (2019) even speculates on reasons for the "historical absence" of double descent.

But double descent is hardly a modern deep learning phenomenon. The original introduction of double descent surprisingly dates back three decades earlier, at least to Opper et al. (1989), and was also presented in Opper et al. (1990), LeCun et al. (1991), and Bös et al. (1993). It can also be understood and reproduced using other model classes. In fact, the Belkin et al. (2019) paper itself demonstrates double descent with random forests and random feature models in addition to two-layer fully-connected neural networks.

Following Maddox et al. (2020), consider Figure 1 (bottom left), showing cross-entropy loss on CIFAR-100 with increases in the width of each layer of a ResNet-18, training to convergence. In the underparametrized regime (yellow), train and test loss both decrease, as increases in flexibility enable the model to capture more useful information content in the data, which increases the effective dimensionality (Section 3.2) of the Hessian of the trained parameters. In the the transition regime (pink), the training loss is still decreasing, and the information content in the trained parameters of the model is still increasing, leading to a continued increase in effective dimensionality. However, the way the model is capturing structure is leading to some overfitting, increasing the test loss. In the interpolation regime (green), where number of parameters exceed the number of data points, test loss again decreases with increases in parameters. Importantly, in this second descent, all models achieve a perfect fit to the data, and therefore increased flexibility *cannot* be the reason that the models with more parameters achieve better generalization. Instead, as we continue to increase the number of parameters, the volume of compressible flat solutions grows, making these solutions more discoverable during training. The effective dimensionality of the solutions thus decreases, and generalization will improve.

In Figure 1 (bottom right) we show double descent for the mean-squared error of a linear model, $Xw = y$, where $X$ is an $n \times d$ matrix of features, $w$ represents $d$ parameters, and $y$ are the $n$ datapoints. This model uses weakly informative features $y + \epsilon$, where $\epsilon \sim \mathcal{N}(0, 1)$. Once $d > n$, the model can interpolate the data perfectly with infinitely many parameter settings $w$. Bartlett et al. (2020) shows that the least squares solution $w^* = (X^\top X)^{-1} X^\top y$ provides the minimum $\ell_2$ norm solution amongst parameter settings that perfectly fit the data, favouring simpler models that rely primarily on the most informative directions in the feature space. We can also consider the effective dimensionality of the parameter covariance matrix (inverse Hessian), or, equivalently, the number of relatively small eigenvalues of

the Hessian $X^\top X$. Using the Marchenko-Pastur distribution in random matrix theory (Marchenko & Pastur, 1967; Dobriban & Wager, 2018), we can predict in this setting that the small eigenvalues will get larger as the number of parameters $d$ increases past $n$, which in turn decreases the variance error (in the bias-variance decomposition) of the model (Hastie et al., 2022), leading to better generalization. Curth et al. (2023) provides a detailed analysis of double descent in classical statistical models, including a generalized notion of effective parameters.

As with benign overfitting, these notions of simplicity can be formally characterized using countable hypothesis or PAC-Bayes bounds. It is also possible to track double descent with formal PAC-Bayes bounds, as in Lotfi et al. (2022a, Figure 7). In the second descent, larger models achieve similar empirical risk, but can be more compressible.

## 7. Alternative Views

The alternative view is that benign overfitting, double descent, and overparametrization, are largely modern deep learning phenomena that require rethinking generalization.

*How did this alternative view (which is quite mainstream!) arise in the first place?*

**The bias-variance trade-off** decomposes expected generalization loss into the expected data fit (the bias) and the expected square difference between fits (the variance), over the data generating distribution. Constrained models tend to have high bias and low variance, and unconstrained models tend to have low bias and high variance, suggesting the "U shaped" curve in the classical regime of double descent. Accordingly, textbooks do indeed warn "a model with zero training error is overfit to the training data and will typically generalize poorly" (Hastie et al., 2017). But "trade-off" is a misnomer: models such as our order-dependent polynomial in Section 2, or ensembles (Bishop, 2006; Wilson & Izmailov, 2020), can have low bias and low variance.

**Rademacher complexity**, which measures the ability for a function class to fit uniform $\pm 1$ labels, will not lead to meaningful generalization bounds for models that perform benign overfitting. Similar reasoning applies to VC and fat-shattering dimensions. But even in the more recent retrospective *"...still requires re-thinking generalization"* (Zhang et al., 2021) there is only a single sentence on PAC-Bayes: "where the learning algorithm is allowed to output a distribution over parameters, new generalization bounds were also derived". As we discussed in Section 3, PAC-Bayes and countable hypothesis bounds can apply to deterministically trained models. They additionally provide a rigorous conceptual understanding of this generalization behaviour, and have existed for many decades. The basic idea behind the bounds is even described in well-known textbooks, for

example Shalev-Shwartz & Ben-David (2014, Chapter 7.3). However, these frameworks must not have been broadly known or internalized, and the deterministic variants as non-vacuous bounds on large networks became more visible somewhat later, for example in Lotfi et al. (2022a).

**The implicit regularization of neural networks** differs, for instance, from our running example of a large polynomial with order-dependent regularization. However, both types of regularization are examples of soft inductive biases, and we have discussed how increasing the size of a neural network can increase its implicit regularization. Moreover, this implicit regularization is reflected in the generalization frameworks of Section 3, and characterized by quantities such as effective dimension. Implicit regularization is also not specific to neural networks, and applies to our random feature linear model in Section 6. Moreover, contrary to conventional wisdom, the implicit regularization of stochastic optimizers is not likely to play a major role in deep learning generalization, as discussed in Section 5. On the other hand, we are still in the early stages of understanding precisely how and why scale and other factors influence the implicit regularization in neural networks.

## 8. Discussion

Overparametrization, benign overfitting, and double descent are intriguing phenomena, worthy of (further) study. However, contrary to widely held beliefs, they are consistent with long-standing frameworks for understanding generalization, reproducible using other model classes, and intuitively understandable. Going forward, we hope we can help bring different communities closer together, so that a variety of perspectives and generalization frameworks are less at risk of being overlooked.

*Grokking* and *scaling laws* are other phenomena of recent interest, similarly fascinating and worth understanding further. But unlike the phenomena we consider in this paper, they are not typically presented as evidence we need to re-think generalization frameworks, or as deep learning phenomena. And indeed, it is being shown that scaling laws and grokking apply to linear models (Lin et al., 2024; Atanasov et al., 2024; Miller et al., 2023; Levi et al., 2023). Importantly, PAC-Bayes and countable hypothesis bounds are also consistent with large LLMs, as we saw in Figure 2, and recent work even shows that these bounds describe Chinchilla scaling laws (Finzi et al., 2025).

If these phenomena aren't distinct to deep neural networks, then what is? Deep learning of course is different from other model classes, and not well understood in many ways. Its empirical success alone sets it apart. In Appendix A, we discuss phenomena that are relatively distinct to neural networks, such as representation learning, mode connectivity, and relatively universal in-context learning.

**Acknowledgements:** We thank Shikai Qiu, Pavel Izmailov, Marc Finzi, Gautam Kamath, Greg Benton, Micah Goldblum, Alan Amin, Jacob Andreas, Alex Alemi, Lucas Beyer, Mikhail Belkin, Sanae Lotfi, Alan Jeffares, Alicia Curth, Martin Marek, Sanyam Kapoor, Patrick Lopatto, Preetum Nakkiran, Thomas Dietterich, and Sadhika Malladi for helpful discussions. This work was supported in part by NSF CAREER IIS-2145492, NSF CDS&E-MSS 2134216, NSF HDR-2118310, BigHat Biosciences, Capital One, and an Amazon Research Award.

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

# A. What is Different or Mysterious?

If these phenomena aren't distinct to deep neural networks, then what is?

Deep neural networks are certainly different from other model classes, and in many ways they are not well understood. Their empirical performance alone sets them apart. Indeed, the substantial disparity in performance between deep convolutional neural networks and the next leading approaches on ImageNet is responsible for renewed interest in (and the subsequence dominance of) this model class (Krizhevsky et al., 2012). But if they are not in fact distinguished by overparametrization, benign overfitting, or double descent, what does make these models different?

To conclude, we briefly highlight some, but surely not all, particularly salient properties and generalization behaviours that are relatively distinctive to neural networks.

## A.1. Representation Learning

*Representation learning* is largely what sets neural networks apart from other model classes. What does representation learning actually mean?

Most model classes can be expressed as an inner product of parameters $w$ and basis functions $\phi$: $f(x, w) = w^\top \phi(x)$. While the function class may be highly flexible (in some cases more so than any neural network we can fit in memory) (Williams & Rasmussen, 2006), and the basis functions non-linear, the basis functions typically are a priori *fixed*. For example, we may be using a polynomial basis, Fourier basis, or radial basis. Beyond possibly a few hyperparameters, such as the width of a radial basis, the basis functions do not typically have many of their own parameters that are learned from data. Neural networks, by contrast, specify an *adaptive* basis: $f(x, w) = w^\top \phi(x, v)$ where $v$ are a relatively large set of parameters to be learned (the weights of the neural network) that significantly control the shape of the basis functions, typically through a hierarchical formulation involving successive matrix multiplications passed through pointwise non-linearities $\sigma$: $f(x, w) = W_{p+1}\sigma(W_p \ldots \sigma(W_2\sigma(W_1 x))\ldots)$. Here, $\phi(x, v) = \sigma(W_p \ldots \sigma(W_2\sigma(W_1 x))\ldots)$, and $v = W_1, \ldots, W_p$.

At first glance, it may seem unnecessary to *learn* basis functions. After all, as we saw in Section 5, we can achieve as much flexibility as we need — universal approximators — with fixed basis functions, through kernels. But by *learning* the basis functions, we are effectively learning the kernel — a similarity metric for our particular problem. Being able to learn a similarity metric is profoundly important for high dimensional natural signals (images, audio, text, . . . ), where standard notions of similarity, such as Euclidean distance, break down. This notion of representation learning as similarity learning transcends the standard basis function view of

modelling. For example, it also applies to procedures such as k-nearest neighbours (knn), where performance hinges on choosing a fixed *distance measure*, which ideally could instead be learned.[1]

To consider a simple example of representation learning, suppose we wish to predict the orientation angle of a face. Faces with similar orientation angles may have very different Euclidean distances of their pixel intensities. But the internal representation of a neural network can learn that, for the task at hand, they should be represented similarly. In other words, the Euclidean distances between *deep layers*, rather than *raw inputs*, for faces with similar orientation angles will be similar. This ability to learn similarity metrics is necessary for *extrapolation* — making predictions far away from the data. Euclidean distances on the raw inputs is perfectly fine if we have enough datapoints distributed densely enough for interpolation to work well: if we have many examples of 59 and 61 degree rotations, interpolation will work reasonably well for predicting a 60 degree rotation. But through representation learning, a neural network will be able to accurately predict a 60 degree rotation from having seen only distant angles (Wilson et al., 2016).

Representation learning, however, is *not* unique to neural networks. It's not uncommon to see claims about what neural networks can do that kernel methods cannot (e.g., Allen-Zhu & Li, 2023). Nearly always these contrasts are implicitly assuming that the kernel is fixed. But in fact *kernel learning* is a rich area of research (Bach et al., 2004; Gönen & Alpaydın, 2011; Wilson & Adams, 2013; Wilson et al., 2016; Belkin et al., 2018; Yang & Hu, 2020). And there is no need to view kernel methods and neural networks as competing. In fact, they are highly complementary. Kernel methods provide a mechanism to use models with an infinite number of basis functions, and neural networks provide a mechanism for adaptive basis functions. There is no reason we cannot have infinitely many adaptive basis functions! *Deep kernel learning* (Wilson et al., 2016) precisely provides this bridge, and was initially demonstrated on the very orientation angle problem we considered here. This approach has recently seen a resurgence of interest for epistemic uncertainty representation that only requires a single forward pass through the network.

Neural networks are also not the only way to do representation learning. In low-dimensional spaces, for example, it can be effective to interpolate on spectral densities (learning the salient frequencies of the data) as a mechanism for kernel learning (Wilson & Adams, 2013; Benton et al., 2019).

But neural networks are a relatively efficient way to learn adaptive basis functions, especially in high dimensions. It's

---

[1] While $k$-nearest neighbours could be derived from a basis function view, it's not the most natural interpretation.

not entirely clear why, either. Not only do neural networks learn a notion of distance, this distance measure changes depending on where we are in input space $x$ — it is *non-stationary*. Non-stationary metric learning is notoriously difficult without making certain assumptions that are well-aligned with data (Wilson & Adams, 2013). Fundamentally, neural networks provide hierarchical representations for data, and these hierarchies are often a natural representation of real-world problems. As we will discuss in the next Section A.2, they also provide a strong bias for low Kolmogorov complexity that could align well with natural data distributions.

## A.2. Universal Learning

Historically, the conventional wisdom is to build specialized learners with assumptions constrained to specific problem settings. For example, if we are modelling molecules, we could hard-code rotation invariance — and talk with domain experts to understand the other constraints we want to impose on our model. This approach is often motivated from the *no free lunch theorems* (Wolpert, 1996; Wolpert & Macready, 1997; Shalev-Shwartz & Ben-David, 2014), which say that every model is equally good in expectation over all datasets drawn uniformly. These theorems typically imply that if a model performs well on one problem, it has to perform poorly on other problems, leading to the desire for highly tailored assumptions.

However, developments in deep learning have run exactly contrary to this conventional wisdom! We have seen a confluence of models — a move from hand-crafted feature engineering (SIFT, HOG, etc.), to neural networks specialized to particular domains (CNNs for vision, RNNs for sequences, MLPs for tabular data, . . . ), to *transformers for everything*. This result can be explained by both neural networks models, and the distribution of naturally occurring data (rather than data sampled uniformly), having a bias for low Kolmogorov complexity. Surprisingly, even models designed for specific domains, such as convolutional neural networks for image recognition, provably have inductive biases for completely different modalities of data, such as tabular data, due to this bias (Goldblum et al., 2024). Starting with a neural network trained on one problem, it is possible to derive non-vacuous generalization bounds for performance on other problems and even other modalities, through upper bounding Kolmogorov complexity.

Indeed, *in-context learning*, the ability for a model to learn without updating its parameters, works distinctly well for neural networks. In some sense many classical models are also performing in-context learning, or something close to it: when we use a Gaussian process with a fixed RBF kernel, condition on some training data, and then sample the posterior predictive, we are doing conditional generation without updating the model representation. But the relative universality of in-context learning for transformers is unprecedented. For example, a standard LLM pre-trained on text completion can surprisingly make competitive zero-shot time series forecasts relative to purpose-built time series models trained on time series data (Gruver et al., 2024)!

In other words, not only are neural networks learning rich representations of data, they are learning representations that are relatively universal across real-world problems, compared to other model classes. We emphasize that for in-context learning, these are not *fixed* representations. During pre-training, transformers learn to learn, discovering inductive principles such as Occam's razor (Gruver et al., 2024; Goldblum et al., 2024). In the GP analogy, we can think of the pre-trained transformer as a large mixture of GP experts, with different kernels. Conditioned on the downstream dataset, the transformer selects the appropriate combination of kernels, based on what it has seen in pre-training.

## A.3. Mode Connectivity

*Mode connectivity* is a surprising phenomenon that is relatively distinct to neural networks (Garipov et al., 2018; Draxler et al., 2018; Frankle et al., 2020; Freeman & Bruna, 2017; Adilova et al., 2023). If we re-train a neural network multiple times with different initializations, it was believed that we would converge to isolated local optima, with significant loss barriers between them. However, it was discovered that there are simple paths between these different solutions that maintain essentially zero training loss, as illustrated in Figure 5 (Garipov et al., 2018; Draxler et al., 2018). In other words, it is a misnomer to even refer to the converged solutions as local optima! Importantly, the parameter settings along mode connecting curves correspond to different functions that will make different predictions on test points, rather than representing degeneracies in the model specification, such as parameter symmetries.

Mode connectivity has profound implications for understanding generalization in deep learning. Indeed, historically one of the most common objections to deep learning is the extreme multimodality of the loss landscapes (training objectives). Mode connectivity shows instead that the solutions that we are finding in practice are all connected together. Accordingly, understanding mode connectivity and developing practical procedures inspired by this phenomenon has become a vibrant area of research (e.g., Kuditipudi et al., 2019; Frankle et al., 2020; Benton et al., 2021; Zhao et al., 2020; Ainsworth et al., 2022).

Mode connectivity has also inspired popular optimization procedures such as stochastic weight averaging (SWA) (Izmailov et al., 2018), which in turn inspired model soups (Wortsman et al., 2022), and the area of model merging (Ainsworth et al., 2022; Yang et al., 2024).

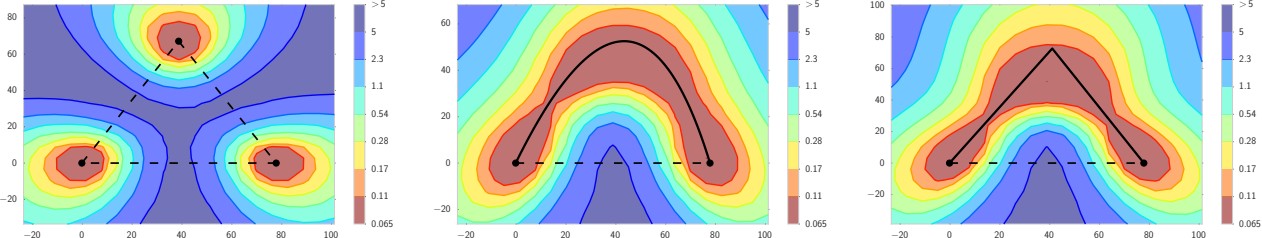

*Figure 5.* **Modes in the neural network landscape are connected along curves.** Three different two-dimensional subspaces of the $\ell_2$-regularized cross-entropy loss landscape of a ResNet-164 on CIFAR-100 as a function of network weights. The horizontal axis remains fixed, anchored to the optima of two independently trained networks, while the vertical axis varies across panels. **Left:** Conventional assumption of isolated optima. **Middle** and **Right:** Alternative planes where optima are connected via simple curves while maintaining near-zero loss. *Mode connectivity* is relatively distinct to deep neural networks. Figure adapted from Garipov et al. (2018).

But, like representation learning, mode connectivity is not entirely unique to neural networks (e.g., Kanoh & Sugiyama, 2024). However, mode connectivity is largely a deep learning phenomenon, clearly only applicable to sophisticated non-convex loss landscapes.

### A.4. Extended Discussion

**What is the role of the optimizer in deep learning generalization?** There is a conventional wisdom that the green and pink colours in Figure 7 are essentially inverted, and that the main reason deep learning works is because the implicit biases of stochastic optimizers cause them to traverse a relatively small subspace of low loss solutions with good generalization. However, it has been shown that full batch gradient descent, and even *guess and check*, stopping when the loss falls below a threshold, can find solutions with similar generalization as stochastic optimization (Geiping et al., 2021; Chiang et al., 2022), in alignment with Figure 7 (right). While in principle it is possible for an optimizer to still find bad optima under such a loss landscape, it would have to be actively adversarial. Far from adversarial, stochastic optimization has biases that can indeed improve generalization. But, importantly, these biases are not *necessary* for respectable generalization. Of course, stochastic optimization is much more computationally practical than the alternatives. No one is suggesting we use guess and check! Moreover, developing optimizers which generalize better under a given computational budget is a particularly exciting research direction, especially with recent results showing the rise of second-order optimizers (Liu et al., 2025; Vyas et al., 2024). Finally, the generalization bounds of Section 3.1 can be evaluated regardless of whether the model uses stochastic optimization, and indeed these bounds track the benign overfitting behaviour of Gaussian processes, which perform Bayesian inference.

**What is the relationship between structural risk minimization and soft inductive biases?** SRM is a way to encode a soft inductive bias, but is more narrowly focused, and often differently motivated. SRM is often used as a mechanism to reduce VC dimension, trading off data fit with model complexity. It is not typically used as a prescription for arbitrarily flexible models, and indeed model selection tools with priors corresponding to standard $\ell_2$ regularization suggest we should use intermediate order models (Bishop, 2006). A key point in this paper is that we can embrace models that fit data perfectly (including noise) but still have a bias for simplicity. Other ways of implementing soft inductive biases include overparametrization, Bayesian priors and marginalization, the optimizer, and architectural specification.

**Can restriction biases have computational benefits?** While we are primarily considering the principles of model construction in an idealized setting, where computation is not a constraint, restriction biases such as sparsity and parameter sharing can be a practical design decision for computational reasons. However, even when considering computation, restriction biases can be undesirable. Recent work shows, for instance, that parameter sharing can be a poor principle for compute-optimal scaling (Potapczynski et al., 2024).

**How can we better understand generalization in deep learning?** There are many fascinating open questions in deep learning generalization. As an approach, we believe it is promising to analyze the solutions neural networks actually reach to explain their behaviours. The generalization bounds of Section 3.1 are fully empirical, non-asymptotic, and can be evaluated using a single sample. We view being able to empirically evaluate the bounds as essential in determining how much of the empirical model behaviour is actually explained by the theory. We have found the Solomonoff prior particularly useful for evaluating descriptive generalization bounds. Solomonoff induction uses a maximally overparametrized model, containing every possible program, but formalizes an ideal learning system that assigns exponentially higher weights to shorter programs. In the future, it would be enlightening to investigate prop-

erties of priors that may lead to tighter bounds, ever more closely describing deep learning generalization. It would be particularly exciting to move beyond Kolmogorov complexity as a measure of information content, in order to distinguish between incompressibility due to randomness versus incompressibility due to structural complexity.

## B. Common Misconceptions about PAC-Bayes

There are several common misconceptions about PAC-Bayes and countable hypothesis bounds.

**Misconception: PAC-Bayes only applies to stochastic networks**, rather than the deterministically trained networks we use in practice, since it characterizes the expected generalization of a posterior sample. However, the posterior need not be the Bayes posterior, and we can evaluate the PAC-Bayes bound with a point-mass posterior and a discrete hypothesis space: using the relative entropy definition of the KL divergence, $\mathbb{KL}(Q \parallel P) = \mathbb{H}(Q, P) - \mathbb{H}(Q)$, the cross-entropy $\mathbb{H}(Q, P)$ becomes $\log_2 \frac{1}{P(h)}$ and the entropy $\mathbb{H}(Q)$ or "surprise" in seeing a sample from a point mass $Q$ is zero, recovering a bound very similar to the countable hypothesis bound. Alternatively, the countable hypothesis bound directly applies to deterministically trained models.

**Misconception: the countable hypothesis bound doesn't apply to models with continuous parameters.** The neural networks we use are in fact programs on a computer, and therefore must represent a finite hypothesis space. The weights can only take a finite number of values determined by the precision, such as floating point. There is a related misconception that the countable hypothesis bounds must then be loose because there are many hypotheses represented by floating point neural network parameter values. However, the form of the bounds makes clear that we should avoid strictly measuring the number of hypotheses and instead understand generalization from the perspective of which hypotheses are a priori likely. Indeed, these bounds can be tighter for larger models representing more hypotheses (Lotfi et al., 2024a).

**Misconception: these bounds become loose as we increase the number of parameters.** While many bounds, including some PAC-Bayes bounds, do have parameter counting terms (Jiang et al., 2019), this is not true of all PAC-Bayes or countable hypothesis bounds. Indeed, recent bounds can become tighter with increasing numbers of model parameters (Lotfi et al., 2022a; 2024a;b) because larger models can have a stronger compression bias, leading to a decreased complexity penalty in the bound.

**Misconception: tight neural network bounds are for unrealistic model compressions.** There is a form of bound, referred to as a *compression bound*, which bounds the generalization of a model whose parameters have been com-

pressed into a lower-dimensional space. It is true that this approach had early success in achieving non-vacuous bounds for larger neural networks on larger datasets (Zhou et al., 2018; Lotfi et al., 2022a). However, there are a few misconceptions to address: (1) the compression techniques used, such as forming linear subspaces of the parameter space, famously perform often nearly as well as the original model (Li et al., 2018). The bounds are often describing a model that is practically compelling, rather than an unrealistic model reduction; (2) the ability to compress larger neural networks into lower dimensional subspaces is informative about generalization; (3) the more recent non-vacuous bounds are not compression bounds, such as the bounds on billion parameter LLMs in Lotfi et al. (2024b) and Finzi et al. (2025).

**Misconception: Kolmogorov complexity is not computable and so generalization bounds based on a Solomonoff prior cannot be evaluated.** The prefix-free Kolmogorov complexity $K(h)$ represents the shortest program in bits to represent $h$ using some pre-specified coding. While we cannot compute the shortest program, we can upper bound the shortest program by the stored filesize of the model and a constant given by terms that do not depend on the data, such as the size of the (e.g., Python) script we use to load and run the model. We can absorb these constant terms that do not depend on the data, represented by $A$, into the Solomonoff prior, by working with $K(h|A)$. We can then in turn upper bound the non prefix-free (standard) Kolmogorov complexity $C$ (conditioned on $A$) by the stored filesize of the trained model to compute informative generalization bounds.

Incidentally, a profound property of Kolmogorov complexity is that it measures the absolute information independently of the programming language or Universal Turing Machine used. We can write a compiler that translates the code of one language to another without reference to any particular strings. In particular, the *invariance theorem* upper bounds the difference in Kolmogorov complexity under any two Universal Turing Machines by the *shortest possible compiler* (Kolmogorov, 1965; Li & Vitányi, 2008). Such a compiler would typically be at most on the order of kilobytes, which is negligible compared to typical ML datasets which can be terabytes.

**Misconception: the bounds only hold if the prior $P(h)$ is not misspecified.** The bound does not require that the prior be used to generate the correct hypothesis, or contain the hypothesis, or even be used by the model we are bounding. It simply provides a mechanism to compute the bound. If, for example, the prior used in the bound favours simple solutions, and the model has a prior that favours complex solutions, we will merely have a looser bound. The assumptions of the bound apply to the models we are us-

ing in practice, including, for instance, the CIFAR benign overfitting experiments of Zhang et al. (2016).

# C. Other Generalization Frameworks

*Rademacher complexity* (Bartlett & Mendelson, 2002) exactly measures the ability for a model to fit uniform $\{+1, -1\}$ random noise. In particular, the Rademacher complexity of a hypothesis space $\mathcal{H}$ and an input sample $\{x_i, \ldots, x_n\}$ is $\mathcal{R}(\mathcal{H}) = \mathbb{E}_\sigma \left[ \sup_{h \in \mathcal{H}} \frac{1}{n} \sum_{i=1}^n \sigma_i h(x_i) \right]$, where $\sigma_i$ are i.i.d. Rademacher random variables ($\{+1, -1\}$ with equal probability). The expected risk of a hypothesis $h$ is then bounded as $R(h) \leq \hat{R}(h) + 2\mathcal{R}(\mathcal{H}) + C$, where $C$ is a constant defined by the loss function, $n$, and the confidence $1 - \delta$ of the bound. Thus, if the model has a hypothesis space $\mathcal{H}$ that can fit the Rademacher noise, then the Rademacher generalization bound will be uninformative — unless it is adapted to include a prior that assigns higher density to certain solutions over others, much like how we move from a standard finite hypothesis bound with a uniform prior over hypotheses to a countable hypothesis bound with arbitrary prior in Appendix D.

Similarly, the *VC dimension* (Vapnik et al., 1994) measures the largest integer $d$ such that the hypothesis space $\mathcal{H}$ can fit ("shatter") any set of $d$ points with $\{+1, -1\}$ labels (e.g., classify these points in all $2^d$ possible ways). If the VC dimension $\mathcal{H}$ is $d$, then the expected generalization error is bounded as $R(h) \leq \hat{R}(h) + \mathcal{O}\left(\sqrt{\frac{d \log(n)}{n}}\right)$. Thus, models with large hypothesis spaces have uninformative VC generalization bounds.

The *fat-shattering dimension* (Alon et al., 1997) $\text{fat}_\gamma(\mathcal{H})$ refines the VC dimension to fitting ("shattering") labels by some margin $\gamma$ (or the function having all possible values within some range $[y_i - \gamma, y_i + \gamma]$ for each target $y_i$). The fat-shattering dimension is closely related to Rademacher complexity: $\mathcal{R}(\mathcal{H}) \leq c\gamma \sqrt{\frac{\text{fat}_\gamma(\mathcal{H})}{n}}$. We can bound expected generalization as $R(h) \leq \hat{R}(h) + \mathcal{O}\left(\sqrt{\frac{\text{fat}_\gamma(\mathcal{H}) \log(n)}{n}}\right)$. With larger $\gamma$, the fat-shattering dimension $d$ will decrease, as the constraints are harder to satisfy. The ability to fit noise, and a flexible hypothesis space, can be explained by the fat-shattering dimension if the model can only fit noise with small but not larger $\gamma$; however, the fat-shattering dimension is in general difficult to compute for arbitrary neural networks.

We provide a comparative summary of different generalization bounds in Table 1.

# D. Countable Hypothesis Bound

**Theorem D.1.** *Consider a bounded risk $R(h, x_i) \in [a, a + \Delta]$ and a countable hypothesis space $h \in \mathcal{H}$ for which we have a prior $P(h)$ that does not depend on $\{x_i\}$. Let the empirical risk $\hat{R}(h) = \frac{1}{n} \sum_{i=1}^n R(h, x_i)$ be a sum over independent random variables $R(h, x_i)$ for a fixed hypothesis $h$. Let $R(h) = \mathbb{E}[\hat{R}(h)]$ be the expected risk.*

*With probability at least $1 - \delta$:*

$$R(h) \leq \hat{R}(h) + \Delta \sqrt{\frac{\log 1/P(h) + \log 1/\delta}{2n}}. \quad (4)$$

*Proof (Lotfi et al., 2024a).* As $n\hat{R}(h)$ is the sum of independent and bounded random variables, we can apply Hoeffding's inequality (Hoeffding, 1994) for a given choice of $h$. For any $t > 0$

$$P(R(h) \geq \hat{R}(h) + t) = P(nR(h) \geq n\hat{R}(h) + nt)$$
$$P(R(h) \geq \hat{R}(h) + t) \leq \exp\left(-2nt^2/\Delta^2\right).$$

We will choose $t(h)$ differently for each hypothesis $h$ according to

$$\exp\left(-2nt(h)^2/\Delta^2\right) = P(h)\delta.$$

Solving for $t(h)$, we have

$$t(h) = \Delta \sqrt{\frac{\log 1/P(h) + \log 1/\delta}{2n}} \quad (5)$$

This bound holds for a fixed hypothesis $h$. However, for an $h^*(\{x\})$ constructed using the training data, the random variable

$$\hat{R}(h^*) = \frac{1}{n} \sum_{i=1}^n R(h^*(\{x\}), x_i),$$

cannot be decomposed as a sum of independent random variables. Since $h^* \in \mathcal{H}$, if we can bound the probability that $R(h) \geq \hat{R}(h) + t(h)$ for *any* $h$, then the bound also holds for $h^*$.

Applying a union over the events $\bigcup_{h \in \mathcal{H}} \left[ R(h) \geq \hat{R}(h) + t(h) \right]$, we have

$$P(R(h^*) \geq \hat{R}(h^*) + t(h^*)) \leq P\left( \bigcup_{h \in \mathcal{H}} \left[ R(h) \geq \hat{R}(h) + t(h) \right] \right)$$
$$\leq \sum_{h \in \mathcal{H}} P\left( R(h) \geq \hat{R}(h) + t(h) \right)$$
$$\leq \sum_{h \in \mathcal{H}} P(h)\delta = \delta.$$

Therefore we conclude that for any $h$ (dependent on $x$ or not), with probability at least $1 - \delta$,

$$R(h) \leq \hat{R}(h) + \Delta \sqrt{\frac{\log 1/P(h) + \log 1/\delta}{2n}}.$$

$\square$

*Table 1.* Summary of Generalization Bounds

| Bound Type | Measure | Generalization Bound | Introduced | Interpretation |
|---|---|---|---|---|
| **Rademacher** | $\mathcal{R}_n(\mathcal{H})$ | $R(h) \le \hat{R}(h) + 2\mathcal{R}_n(\mathcal{H}) + 3\sqrt{\frac{\log(2/\delta)}{2n}}$ | 2000s | Measures expected maximum correlation any $h \in \mathcal{H}$ can achieve with uniform $\{+1, -1\}$ samples. Does not explain overparametrization, benign overfitting, or double descent. |
| **VC Dimension** | $d$ | $R(h) \le \hat{R}(h) + \mathcal{O}\left(\sqrt{\frac{d\log(n)}{n}}\right)$ | 1990s | Measures number $d$ uniform $\{+1, -1\}$ samples any $h$ can fit. Does not explain overparametrization, benign overfitting, or double descent. |
| **Fat Shattering** | $\text{fat}_\gamma(\mathcal{H})$ | $R(h) \le \hat{R}(h) + \mathcal{O}\left(\sqrt{\frac{\text{fat}_\gamma(\mathcal{H})\log(n)}{n}}\right)$ | 1990s | Refines VC for real-valued functions and margin $\gamma$. Possibly describes benign overfitting for larger $\gamma$, but can be hard to evaluate. |
| **PAC-Bayes** | $\text{KL}(Q\|P)$ | $R(h) \le \hat{R}(h) + \mathcal{O}\left(\sqrt{\frac{\text{KL}(Q\|P)+\log(n/\delta)+2}{2n-1}}\right)$ | 1990s | Generalization is controlled by which solutions are likely under the prior, rather than size of the hypothesis space. Describes overparametrization, benign overfitting, and double descent. |
| **Finite Hypothesis** | P(h) | $R(h) \le \hat{R}(h) + \mathcal{O}\left(\sqrt{\frac{\log 1/P(h)+\log 1/\delta}{2n}}\right)$ | 1980s | Generalization is controlled by which solutions are likely under the prior. Applies to deterministic models. Prior can be evaluated through bound on Kolmogorov complexity given by storage space of trained model. Non-vacuous bounds for million and billion parameter neural nets. Bounds often *improve* for larger models. Describes overparametrization, benign overfitting, double descent. |

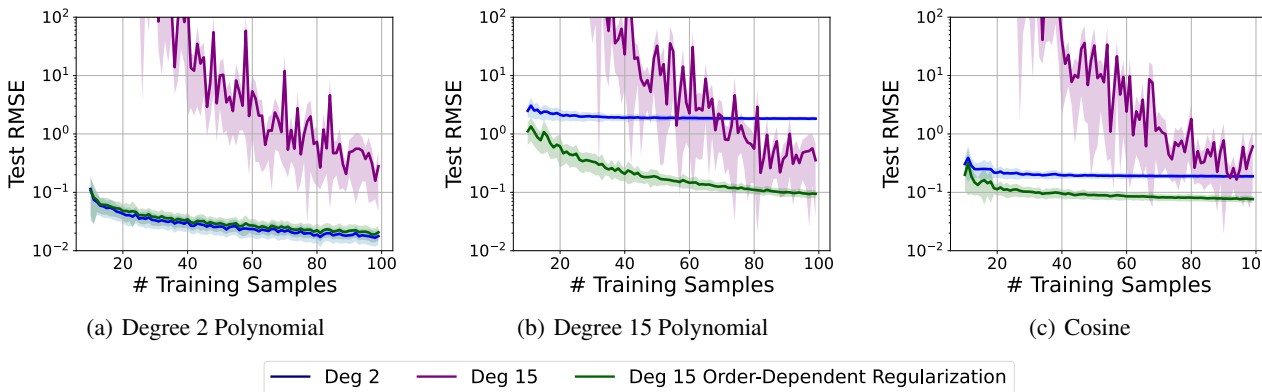

(a) Degree 2 Polynomial      (b) Degree 15 Polynomial      (c) Cosine

— Deg 2    — Deg 15    — Deg 15 Order-Dependent Regularization

*Figure 6.* **Flexibility with a simplicity bias can be appropriate for varying data sizes and complexities.** We use 2nd, 15th, and regularized 15th order polynomials to fit three regression problems with varying training data sizes, generated from the functions described in (a)-(c). We use a special regularization penalty that increases with the order of the polynomial coefficient. We show the average performance $\pm$ 1 standard deviation over 100 fits of 100 test samples. By increasing complexity only as needed to fit the data, the regularized 15th order polynomial is as good or better than all other models for all data sizes and problems of varying complexity.

## E. Supplemental Figure on Overparametrization

In Figure 7 we should how increasing the number of model parameters both increases the size of the hypothesis space, and a soft bias for simpler solutions that will provide good generalization, by increasing their relative volume

## F. Experimental Details

In Figure 1(a)(b)(c), we use a $150th$ order polynomial with order-dependent regularization $\sum_j 2^j w_j^2$ (green) to fit regression data generated from (a) $\sin(x)\cos(x^2)$, (b) $x + \cos(\pi x)$, (c) $\mathcal{N}(0, 1)$ noise.

Figure 1(d)(e) is adapted from Wilson & Izmailov (2020), which uses a Gaussian process with an RBF kernel, and

a PreResNet-20 and isotropic prior $p(w) = \mathcal{N}(0, \alpha^2 I)$ and Laplace marginal likelihood, and in turn replicates the CIFAR-10 noisy label experiment in Zhang et al. (2016).

Figure 1(f) is adapted from Maddox et al. (2020) and uses a ResNet-18 with increasing layer width, measures train loss, test loss, and effective dimensionality for $\alpha = 1$. Similar to Maddox et al. (2020), in Figure 1(g) we use the random feature least squares model $Xw = y$ with each column of $X_i = y_i + \epsilon$ where $\epsilon \sim \mathcal{N}(0, 1)$. We measure MSE, and use $\alpha = 10$ to compute the effective dimensionality of the parameter covariance matrix (inverse Hessian).

Figure 2 is adapted from Lotfi et al. (2024a), and evaluates the countable-hypothesis bounds with upper bound on Kolmogorov complexity in Section 3 for LLMs of various sizes.

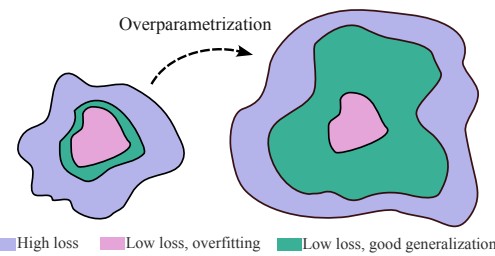

Overparametrization

High loss   Low loss, overfitting   Low loss, good generalization

*Figure 7.* **Increasing parameters improves generalization.** By increasing the number of parameters, flat solutions, which typically provide simpler compressible explanations of the data, occupy a greater relative volume of the total hypothesis space — leading to an implicit soft inductive bias for these simple solutions. Even though overparametrized models often represent many hypotheses (e.g., parameter settings) that overfit the data, they can represent many more that fit the data well and provide good generalization. Overparametrization can simultaneously increase the size of the hypothesis space, and the bias for simple solutions.

Figure 6 fits two $15th$ order polynomials and one $2nd$ order polynomial to data generated from a $2nd$ order polynomial, $15th$ order polynomial, and $\cos(\frac{3}{2}\pi x)$. One of the $15th$ order polynomials uses the order-dependent regularization $\sum_j 0.01^2 j^2 w_j^2$. Train and test input locations are sampled from $\mathcal{N}(0, 1)$. The number of test samples is 100 and the number of train samples range from 10 to 100. For each train sample size, we re-generate data 100 times, and record the RMSE and its standard deviation (represented by shade). A similar result was shown in Goldblum et al. (2024).

All other figures are conceptual figures.

