# OpenReview forum: "Position: Deep Learning is Not So Mysterious or Different"
_ICML.cc/2025/Position_Paper_Track — ICML 2025 Position Paper Track spotlightposter_

### Official Review · Reviewer_S93Y · 2025-02-24

**Significance:** 4
**Argument Clarity:** 3
**Rating:** 4
**Confidence:** 4

**Questions:**

- Regarding double descent, how do you explain the reduction in generalization in the intermediate phase (where the test error increases) under the scope of this paper? More specifically, why does the number of effective parameters/ complexity increases there? Perhaps there is an answer in the paper, but I couldn't find it.

**Discussion Potential:**

3

**Paper Summary:**

The paper attempts to show that several deep learning phenomena (double descent, benign overfitting, and overparameterization) are not unique to deep learning. These phenomena can be classified as applying, what the authors coin as soft inductive biases. Namely, using highly expressive models with implicit or explicit regularization that favor particular solutions that are consistent with the data.  They further claim that these behaviors can be understood through the lenses of PAC-Bayes or finite generalization bound. Since these frameworks focus on which hypotheses are more likely, they can explain and issue non-vacuous generalization bounds even on large models having millions and billions of parameters. The paper addresses each phenomenon separately and shows (or references in the literature) similar behaviors on simpler, nondeep learning, models.

**Position:**

Yes

**Position In Title:**

Yes

**Related Work:**

3

**Strengths And Weaknesses:**

**Strengths**
- I agree with the claim that the number of parameters is not an adequate measure for assessing the model complexity. Looking at the number of effective parameters or Kolmogorov complexity appears more natural.
- The paper is written clearly and has a coherent message.
- The authors provide empirical evidence to their claims, either by using simple models or by providing relevant references in the literature.
- The paper addresses other phenomena that are unique to deep learning.

**Weakness/Questions**
- At the end of Section 2, if I understand correctly, the authors argue against restriction biases and in favor of soft inductive biases for capturing symmetries. Indeed as the authors state CNNs are a canonical example, and the authors present a naive alternative for FC networks. But, to present a complete picture here perhaps it may be worthwhile to discuss other aspects as well, such as parameter efficiency (which can be very important in low-resource environments), or the difficulty in converging to a solution that approximately obeys the symmetry constraint.
- In section 3, it is not clear to me how you arrived at the first inequality under equation (1). It seems to have arrived from the Solomonoff prior, but unless I am missing something, the sign should be reversed. If indeed I am correct, how does it affect some of the claims related to PAC-Bayes?
Also, how did you arrive at the second inequality at the same line? I understand that it originates from Kolmogrov complexity theory, but more background is required here in my opinion.
- Missing reference [1] for GP kernels corresponding to multi-layer neural networks which appeared in parallel to (Lee et al., 2017). Also, [2] may be relevant for non-vacuous PAC-Bayes bounds for deterministic NNs.
- The sum in $\hat{R}$ should be over $n$ instead of $m$

Overall the paper is well written and provide valid explanations for several deep learning phenomena, but if Indeed there is mistake in the bound (my second bullet) it may have a large impact on some of the claims made in this paper. Hence, currently I cannot pass it until further clarifications from the authors.

[1] Matthews, A. G. D. G., Hron, J., Rowland, M., Turner, R. E., & Ghahramani, Z. (2018, February). Gaussian Process Behaviour in Wide Deep Neural Networks. In International Conference on Learning Representations.
[2] Achituve, I., Shamsian, A., Navon, A., Chechik, G., & Fetaya, E. (2021). Personalized federated learning with Gaussian processes. Advances in Neural Information Processing Systems, 34, 8392-8406.

**Support:**

4

---

> ### Author Rebuttal · Authors · 2025-03-28
>
> Thank you for your thoughtful review! Our responses to your questions follow.
>
> _"How do you arrive at the first inequality under equation (1)?"_
>
> The inequality is correct, and the resulting bound valid. But there is indeed a typo in the Solomonoff prior, which fortunately has no effect on the paper. The Solomonoff prior is $P(h) = 2^{-K(h|M)}/Z$, where $Z \leq 1$, since $\sum 2^{-K} \leq 1$ by the Kraft inequality. Therefore $P(h) \geq 2^{-K(h|M)}$ (not less than $2^{-K(h|M)}$ as had been written inline), from which the inequality below equation (1) follows: $P(h) \geq 2^{-K(h|M)}$ $\implies$ $\log(1/P(h)) \leq K(h|M) \log(2)$. The second inequality comes from the fact that you can convert from a prefix-free code (of length K(h) bits) to a non-prefix free code of no more than $C(h)+2\log_2 C(h)$ bits. This is a standard identity in algorithmic information theory (e.g., found in Introduction to Kolmogorov Complexity and Its Applications by Li and Vitanyi, 4th Ed., Ch. 3). It’s mostly bookkeeping, but the basic intuition is to write the binary representation of C(h) and then prepend a description of the length of this binary representation in a self-delimiting form. So we get $K(h|M) \log(2) \leq C(h) \log(2) + 2\log(2)\log_2 C(h)$ $\implies$ $K(h|M) \log(2) \leq C(h) \log(2) + 2\log C(h)$, where $\log$ is the natural logarithm. The prefix-free Kolmogorov complexity has useful mathematical properties (such as satisfying Kraft’s inequality, used in specifying the Solomonoff prior), but the plain Kolmogorov complexity is more convenient in practice for computing generalization upper bounds. We also note this inequality is commonly used in PAC-Bayes papers (e.g., Lotfi et. al, NeurIPS 2022, arXiv 2211.13609, page 4, “universal prior”).
>
> Thanks for catching this typo. To be honest there was a bit of a scare when first seeing your review, but fortunately the typo in specifying the Solomonoff prior does not have any effect on the paper, as the resulting inequality used in the bound is correct! We will fix this typo, and also add a more gentle derivation of the inequality. While a somewhat standard result (e.g., Shalev-Shwartz and Ben-David, Understanding Machine Learning, 2014), we will additionally add the full proof of the main bound in equation (1). It's actually quite pleasingly simple, and instructive (a combination of Hoeffding’s inequality and a union bound).
>
>
> _"Reference [1] for GP kernels corresponding to multi-layer neural networks which appeared in parallel to (Lee et al., 2017)"_
>
> We’re more than happy to add this reference! Thanks for the note. In citing Lee et. al, we just wanted to note that there had been work on multi-layer neural net limits, rather than to assign historical priority. But Matthews et. al is indeed a great foundational paper and does appear to be concurrent with Lee et al., 2017.
>
>
> _"It may be worthwhile to discuss other aspects as well, such as parameter efficiency (which can be very important in low-resource environments), or the difficulty in converging to a solution that approximately obeys the symmetry constraint."_
>
> Indeed these are interesting considerations, and we will add some comments. Interestingly, while convolutional layers lead to parameter efficiency, it has recently been found that parameter-sharing can be a bad principle for compute-optimal scaling (e.g., “Searching for Efficient Linear Layers over a Continuous Space of Structured Matrices”). It is similarly fascinating that even when the symmetry is exact, optimization of a less constrained (e.g., approximately equivariant) solution can be easier.
>
> Thanks again for your thoughtful review. We would appreciate it if you could raise your score, especially in light of our clarifications around the Solomonoff prior and the resulting inequality being valid in our bound. We believe our paper is making a strong and timely contribution, by providing a unifying intuition paired with rigorous generalization frameworks, to jointly understand many phenomena that have been historically presented as mysterious. These frameworks have largely been overlooked for broadly understanding generalization phenomena, which makes our contribution particularly significant and timely. We think with the proposed updates, inspired by your feedback, the paper will be further strengthened.

---

> > ### Comment · Reviewer_S93Y · 2025-04-04
> >
> > I am sorry for the late reply, I wrote a comment a couple of days ago but it wasn't visible to you. In any case, thank you for the answers, and I apologize for any slight concern my original comment may have caused. Most of the points made by me and other reviewers were properly addressed; hence, I raised my score to 4. I would appreciate it if the authors could also address my question in the 'Questions' section of the review.

---

> > > ### Author Response · Authors · 2025-04-04
> > >
> > > No worries. Thanks for your response, and your support! Regarding double descent, in the region where the test error is increasing, the effective dimensionality is still going up, as the training loss is still going down and the model is capturing more structure in the data. The way in which it is capturing this structure is leading to some overfitting which was not possible with smaller scale models in the first descent. This overfitting can be alleviated with Bayesian model averaging and ensembling (e.g., Figure 8 of arXiv 2002.08791). Once the training loss is near-zero, and we start the second descent, all of the models are capable of fitting the data perfectly, and so all that is changing as we increase model size is the soft bias of the model towards simpler solutions that have lower effective dimensionality and generalize better. The bias towards simpler models arises due to the relatively increasing volume of flat solutions in the loss landscape as we increase model size. We will add this clarification to explain the region of increasing test error in double descent. Thanks again for your questions.

---

### Official Review · Reviewer_4rho · 2025-03-07

**Significance:** 3
**Argument Clarity:** 4
**Rating:** 5
**Confidence:** 3

**Questions:**

1. Out of curiosity: What do you think is the reason that the connections you have made to existing generalization frameworks have been overlooked so far?

**Discussion Potential:**

4

**Paper Summary:**

The paper challenges the common narrative that deep learning is different from other classes of machine learning models in terms of generalization. In particular, it takes the position that several phenomena that are said to be unique to deep learning, including benign overfitting and double decent, are neither specific to deep learning nor difficult to explain. Instead, it is shown that existing frameworks such as PAC-Bayes can explain these phenomena, and it is proposed that they can be unified with the concept of *soft inductive biases*.

## update after rebuttal
I will leave my rating as it is. The paper was of high quality at the outset and the authors properly addressed all remaining comments, in particular the concern expressed by reviewer S93Y.

**Position:**

Yes

**Position In Title:**

Yes

**Related Work:**

3

**Strengths And Weaknesses:**

**Strength**

- The paper is very relevant. A conceptual understanding of deep learning is a one of the most pressing problems in ML research. The paper takes a very refreshing position on this issue, challenging common perspectives. The proposed concept of soft inductive biases is very convincing. I think it has high potential to stimulate discussion.

- The argumentation is compelling, well grounded in relevant literature, and clearly illustrated and supported by well-chosen examples (in particular, the running example of a polynomial with order-dependent regularization).

- Overall, the paper is very well written and has a concise and easy to follow structure. I am particularly impressed with how the paper connects its general ideas with specific examples.

- The paper's broad perspective, especially in drawing connections to a variety of generalization frameworks, is much appreciated.

**Weaknesses**

The following are just minor issues and formalities.

- p4, left column: “After exposure to only a very small amount of data, **the soft the bias** would converge to near-perfect rotation equivariance […]"

- p7, left column: “There is often **a a** perceived tension between flexibility and inductive biases”

- This may be a bit pedantic, but I think the reference list is inconsistent and does not match the otherwise high formal quality of the paper. For example, in many cases words are not capitalized where they should be, especially in article titles or journal names. Example:
  - Guedj, B. A primer on **pac-b**ayesian learning. arXiv preprint arXiv:1901.05353, 2019.
  - *In Advances in Neural Information Processing Systems, 2005.* vs *Advances in neural information processing systems, 25, 2012.*

- Also, I would strongly encourage the inclusion of DOI links or URLs where available.

- Figure 4 should be moved to the main body of the manuscript.

- Very minor: when I first read it, I thought the sentence “The textbooks must be re-written!” was a quote from a (non-cited) reference. I would suggest not putting it in quotation marks to make it clear that it is the author's own statement.

**Support:**

4

---

> ### Author Rebuttal · Authors · 2025-03-28
>
> Thanks for your thoughtful, detailed, and supportive review! We really appreciate it. We'll address the editorial comments. Regarding existing generalization frameworks being overlooked, it's honestly a bit puzzling. Our impression is that PAC-Bayes has somewhat broad "name recognition", but it hasn't been more than superficially understood or internalized as a perspective on generalization that can enable an arbitrarily flexible hypothesis space or resolve these phenomena. Perhaps this is because it is erroneously associated with being specific to stochastic or Bayesian models, when it can apply to deterministically trained models, as we highlight in the paper. Moreover, although "finite hypothesis bounds" are well-known, a countable hypothesis bound with a prior is less well-known, and can inspire similar confusion to PAC-Bayes. For example, it is often erroneously assumed that the "prior" in the finite/countable hypothesis bound has to be "well-specified" or used by the model in some way. But this prior is simply a weighting function. If it's not a good description of the model's bias (implicit or explicit), then the bound will simply be loose.
>
> Hopefully our paper can bring about an important change in the way that these frameworks are understood!

---

### Official Review · Reviewer_6dvG · 2025-03-16

**Significance:** 4
**Argument Clarity:** 4
**Rating:** 4
**Confidence:** 3

**Questions:**

- What is the relationship between soft inductive bias and PAC-bayes?
- Appendix A.2: typo? - SWIFT -> SIFT

**Discussion Potential:**

3

**Paper Summary:**

The paper argues that existing generalization theories focused on deep learning, e.g., benign overfitting, double descent, etc., often considered as competing against classical theories, are not phenomena tailored for deep learning, and can be reproduced on simpler models such as linear models or Gaussian process. The paper points out that the form of soft inductive bias, that current deep learning usually adopt, is a distinctive design that makes a mismatch with the prior intuition, given that classical theories have been often explained with hard inductive biases rather than the soft, e.g., in popular books. The paper also remarks that classical theories such as PAC-Bayes bounds can also accurately measure the generalization of large deep learning models, e.g., LLMs, mentioning Lotfi et al. (2022a; b).

**Position:**

Yes

**Position In Title:**

Yes

**Related Work:**

3

**Strengths And Weaknesses:**

**Strengths**

- The paper is notably well-written; it is well-organized, providing clear presentation and evidences.
- The proposed positions are overall clear and solid.
- The claims are well-supported with proper empirical results.
- The paper provides a good overview of existing generalization theories.

**Weakness**

- Although the paper mentions the following: “… We are also not claiming to be the first to note that any of these phenomena can be reproduced using other model classes. …”, I think it would make the paper clearer if the paper could provide a summary of contributions or new observations made in this work, if any.
- I slightly feel that the two discussions about soft inductive biases and PAC-bayes bound look somewhat distinct, and think a more context would be helpful on how the two are related.
- Otherwise, I do not see any major weaknesses, and enjoyed the reading overall.

**Support:**

4

---

> ### Author Rebuttal · Authors · 2025-03-28
>
> We greatly appreciate your thoughtful and supportive review! We respond to your questions below.
>
> _"On the connection between soft inductive biases and PAC-Bayes bounds."_
>
> Soft inductive biases make it possible to embrace an arbitrarily flexible hypothesis space, and still achieve good generalization, by having a soft preference for certain solutions over others, even if they fit the data equally well. Soft inductive biases explain intuitively how overparametrization, benign overfitting, and double descent are possible; in double descent, for example, all the models in the second descent fit the training data perfectly, but the larger models generalize better --- this better generalization is not through flexibility, then, but through a simplicity bias induced by scale.
>
> PAC-Bayes rigorously formalizes the perspective of generalization through soft inductive biases. Unlike other generalization frameworks, which measure the size of the hypothesis space (Rademacher complexity, VC dimension, etc.), suggesting a prescription for _restriction biases_, PAC-Bayes does not penalize a flexible hypothesis space, and instead depends on the soft biases of the model, represented through the prior density. If certain solutions are preferred by the model over others, independently of how well the model fits the data, for example because they are more compressible, then we can provide provable generalization guarantees.
>
> Solomonoff induction is a great illustrative example of the close link between PAC-Bayes and soft inductive biases. Solomonoff induction uses a maximally overparametrized model, containing every possible program, but formalizes an ideal learning system that assigns exponentially higher soft weights to shorter programs. By using PAC-Bayes and countable hypothesis bounds with the Solomonoff prior, we hope to highlight this connection.
>
> We appreciate the question and will clarify these points in the camera ready!
>
> _"Summary of contributions"_
>
> While the ICML call for position papers says "submissions to the position paper track will be judged primarily on whether they present a compelling position that warrants greater exposure within the machine learning community" rather than original research or novel results, our paper does indeed make many contributions. Amongst these, we would highlight: (1) a unifying perspective of soft biases and PAC-Bayes that can simultaneously explain many generalization phenomena at once, including benign overfitting, double descent, and overparametrization; (2) analyzing the solutions that neural networks actually reach to explain their generalization behaviours; (3) a clear response to the tests set forth in “...still requires rethinking generalization” in terms of showing there are generalization measures that behave differently in the presence of signal versus noise for models that can perfectly fit either, even for deterministically trained models; (4) particularly simple examples, such as polynomials with order-dependent regularization, which display generalization behaviour that has been seen as mysterious and specific to deep learning, paired with an understanding of how this behaviour is possible through the lens of soft inductive biases; (5) explanations of these phenomena that do not depend on optimization dynamics; (6) bringing attention to historical frameworks that have been critically overlooked for understanding generalization and explaining a range of phenomena.
>
> Thanks again for your supportive review! We will make updates to the camera ready based on your feedback.

---

### Decision · Program_Chairs · 2025-04-30

**Decision:**

Accept (spotlight poster)

**Comment:**

I recommend accepting this submission.
The reviewers praised the paper's clear and well-structured writing, highlighting its arguments supported by empirical evidence. The paper's broad perspective in summarizing different generalization frameworks (and their relative merits/shortcomings) was seen as valuable.
Reviewers appreciated the focus on an important topic — understanding deep learning generalization. I agree with the reviewers that this paper will generate productive discussion in the community.

My only reason for not recommending a Strong Accept is: the core message of this paper has already been appreciated by parts of the ML theory community for several years now. For example, the long line of theory on Benign Overfitting was initiated by the observation that these so-called “Deep-Learning-specific” phenomena can be reproduced and understood in linear overparameterized models. To quote the introduction of [Bartlett et al 2020], which is now 5 years old: “Thus, to arrive at a scientiﬁc understanding of the success of deep learning methods, it is a central challenge to understand the performance of prediction rules that ﬁt the training data perfectly. In this paper, we consider perhaps the simplest setting where we might hope to witness this phenomenon: linear regression.”